# The Whole-Genome Sequencing and Probiotic Profiling of *Lactobacillus reuteri* Strain TPC32 Isolated from Tibetan Pig

**DOI:** 10.3390/nu16121900

**Published:** 2024-06-16

**Authors:** Qinghui Kong, Zhenda Shang, Shah Nawaz, Suozhu Liu, Jiakui Li

**Affiliations:** 1College of Animal Science, Xizang Agricultural and Animal Husbandry University, Nyingchi 860000, China; kqh610@sina.com (Q.K.); shangzhenda1988@163.com (Z.S.); 2College of Veterinary Medicine, Huazhong Agricultural University, Wuhan 430070, China; malikshahnawaz786@gmail.com; 3Xizang Plateau Feed Processing Engineering Research Center, Nyingchi 860000, China

**Keywords:** *Lactobacillus reuteri*, whole-genome sequence, probiotic characteristics, intestinal health

## Abstract

Gut microbiota are the microbial organisms that play a pivotal role in intestinal health and during disease conditions. Keeping in view the characteristic functions of gut microbiota, in this study, *Lactobacillus reuteri* TPC32 (*L. reuteri* TPC32) was isolated and identified, and its whole genome was analyzed by the Illumina MiSeq sequencing platform. The results revealed that *L. reuteri* TPC32 had high resistance against acid and bile salts with fine in vitro antibacterial ability. Accordingly, a genome sequence of *L. reuteri* TPC32 has a total length of 2,214,495 base pairs with a guanine–cytosine content of 38.81%. Based on metabolic annotation, out of 2,212 protein-encoding genes, 118 and 101 were annotated to carbohydrate metabolism and metabolism of cofactors and vitamins, respectively. Similarly, drug-resistance and virulence genes were annotated using the comprehensive antibiotic research database (CARD) and the virulence factor database (VFDB), in which *vatE* and *tetW* drug-resistance genes were annotated in *L. reuteri* TPC32, while virulence genes are not annotated. The early prevention of *L. reuteri* TPC32 reduced the *Salmonella typhimurium* (*S. Typhimurium*) infection in mice. The results show that *L. reuteri* TPC32 could improve the serum IgM, decrease the intestinal cytokine secretion to relieve intestinal cytokine storm, reinforce the intestinal biochemical barrier function by elevating the sIgA expression, and strengthen the intestinal physical barrier function. Simultaneously, based on the 16S rRNA analysis, the *L. reuteri* TPC32 results affect the recovery of intestinal microbiota from disease conditions and promote the multiplication of beneficial bacteria. These results provide new insights into the biological functions and therapeutic potential of *L. reuteri* TPC32 for treating intestinal inflammation.

## 1. Introduction

Lactic acid bacteria (LAB) are a group of Gram-positive, lactic acid-producing Firmicutes and a well-documented probiotic used in food and dietary supplements since 1990 [1]. LAB regulate animal health in multiple ways. An important aspect of animal host microecology regulation is that LAB can adhere to the intestine, produce organic acids, lower the pH of the intestinal environment, and suppress pathogenic bacterial growth and reproduction [2]. Additionally, LAB play a crucial role in maintaining intestinal mucosa integrity to balance the ecological environment and activate the mucosal immune response to enhance host immunity [3]. *L. reuteri* is the dominant microbiota in animals’ intestines [4]. It functions as a microecological supplement that promotes the growth of cells in the intestinal lining by activating the Wnt/beta-catenin pathway and repairing the damaged intestinal tissue [5,6].

*Salmonella enterica*, a Gram-negative foodborne pathogen, is one of the leading causes of gastroenteritis in humans and animals. Up to now, various strategies, including sanitary barriers, vaccinations [7], bacteriophages [8], antimicrobial peptides [9], probiotics [10], and traditional Chinese medicine [11], have been investigated for *Salmonella* control and treatment in animals. It is worth noting that probiotics decrease the ability of harmful bacteria to survive and multiply in the intestines. This is achieved through various mechanisms, such as enhancing the integrity of the intestinal barrier, outcompeting pathogens, producing substances that kill bacteria, and influencing the structure and function of the intestinal lining [12]. Thus, considering the beneficial characteristics of LAB, our research focused on extracting a strain of *L. reuteri* TPC32 from the feces of healthy Tibetan pigs. The current study investigated the biological properties of *L. reuteri* by acid/bile salt tolerance and antibacterial susceptibility. Moreover, the safety of *L. reuteri* TPC32 was evaluated using hemolytic analysis and oral toxicity testing in mice. The objective of the present investigation was to assess the safety of the isolated strains and determine their potential to reduce the occurrence of bacterial illnesses in a mouse model. Additionally, there is no denying that the diverse range of functions exhibited by *L. reuteri* strains indicates significant heterogeneity in the type of genomes they contain. Thus, genome-level analysis is necessary to precisely comprehend and differentiate the traits of these microorganisms’ strains to a certain extent. Therefore, *L. reuteri* TPC32 whole-genome sequencing and genomic characterization were carried out to further our understanding of this strain’s genome diversity and molecular evolution. A comparative phylogenetic tree was built to determine the evolutionary relationship between *L. reuteri* TPC32 and other *Lactobacillus* strains. The findings of this study have the potential to create new avenues for managing diarrhea, which is linked to bacterial infections in Tibetan pigs.

## 2. Materials and Methods

### 2.1. Isolation and Characterization of L. reuteri TPC32

#### 2.1.1. Isolation of *L. reuteri* TPC32

Fecal samples were obtained randomly from Tibetan pigs of various ages and genders in the Nyingchi district of Tibet, China. The healthy Tibetan pigs were not given antibiotics or probiotics and were fed a regular diet. Sterile cotton swabs and collecting tubes were utilized to gather recently excreted pig feces. The samples were promptly refrigerated, transferred to the laboratory, and processed within a 3 h timeframe from collection. Since the present investigation just utilized fecal samples, ethical review and approval were unnecessary for the animal study.

Briefly, each sample was under aseptic conditions weighing 0.1 g, and sterile phosphate-buffered saline (PBS: pH 7.2; containing L-cysteine at 0.1%) was used for 10-fold gradient dilution until it attained 10^−8^ dilution. Subsequently, 100 μL of each serial dilution was spread onto modified MRS agar plates (MRS purchased from Beijing Land Bridge Technology Company Ltd., Beijing, China, containing CaCO_3_ at 0.4%) and incubated for 48 h at 37 °C under anaerobic conditions. Based on the colony morphology of *Lactobacillus* and the calcium-dissolving zone, a single colony was transplanted into MRS agar plates at 37 °C anaerobically for 48 h [13], and four colonies were selected and cultivated for three generations under the abovementioned conditions. Based on colony morphology, Gram-staining, and catalase tests, the isolates were identified and designated as *Lactobacillus*. Finally, all isolated strains were confirmed and identified by genetic analysis using 16S rRNA amplification with universal primers 27F (5′-AGAGTTTGATCCTGGCTCAG-3′) and 1492R (5′-TACGGCTACCTTGTTACGACTT-3′) [14]. Subsequently, the PCR products were sequenced by the Qingke Biotech Company (Wuhan, China) and subjected to BLAST analysis on the NCBI website.

#### 2.1.2. Acid and Bile Tolerance Properties

The isolated strains were introduced (1% *v*/*v*) into sterile MRS broth that had been initially adjusted to various pH levels (2, 3, 4, and 5) using 0.1 N hydrochloric acid. The control broth did not contain HCl. Subsequently, the survival rate was determined using the viable plate counting method after incubating at 37 °C for 3 h [15]. To assess the viability of the strains in various concentrations of pig bile salt, the strains were cultivated for 24 h and uniformly distributed on MRS agar plates with pig bile salt (Solarbio, Beijing, China) at concentrations of 0.1%, 0.2%, 0.3%, 0.4%, and 0.5%. Unsalted MRS agar plates were used as control samples. The viable colonies were enumerated after 48 h [16].

#### 2.1.3. In vitro Bacteriostatic Test

Bacteriocins were isolated and purified from the *Lactobacillus* extract using a previously published method [17]. The isolated strains were incubated at 37 °C for 24 h in MRS broth. The *Lactobacillus* culture supernatant was centrifuged at 12,000 r/min, 4 °C, for 20 min to collect the clarified supernatant. Later, it was passed through a 0.2 μm filter and stored at −20 °C. Moreover, an inhibition zone assay was performed using the agar diffusion method (the Oxford cup method). Throughout the experiment, 150 μL samples of bacteriocin of *Lactobacillus* were added into an Oxford cup, which was put on the surface of the LB agar medium covered with *S. Typhimurium* (China Medical Bacteria Preservation Management Center 50115) and *Escherichia coli* (*E. coli*, China Medical Bacteria Preservation Management Center 44102) and cultured at 37 °C for 18 h after leaving it for 3 h at room temperature; then, the antibiotic activity was evaluated.

#### 2.1.4. Measurement of the Growth Curve of Isolated Strains

All isolated strains were cultured in MRS broth for 24 h, and then the bacterial solutions (1 × 10^8^ CFU/mL) were inoculated at 5% into MRS broth. Afterward, *OD*_600_ was measured every 2 h up to 36 h (*n* = 3).

### 2.2. The Safety Assessment of L. reuteri TPC32

The bacterial solutions were streaked onto blood agar plates. The hemolytic activity of the strains was evaluated after 24 h at 37 °C, and *Staphylococcus aureus* (*S. aureus*) (China Medical Bacteria Preservation Management Center 26003) was used as a positive control. The Ethics Committee for Animal Research at Xizang Agricultural and Animal Husbandry University, China, approved the study protocol. Female mice weighing 20 ± 2 g were procured from the Lhasa Biopharmaceutical Factory in Lhasa, China. They were kept in a new environment for a week (acclimatization period), with a 12 h light–dark cycle, a temperature of 20 ± 2 °C, and relative humidity of 50 ± 2%. In summary, ten mice were randomly assigned to two groups: The vehicle control group was gavaged with an identical volume of vehicle by force, and the *L. reuteri* TPC32 group was given *L. reuteri* TPC32 at a rate of 1 × 10^8^ CFU/day for fifteen days [18]. During the experiment, mice were observed for general health, behavior, hair shine, and clinical symptoms. All of the animals in each group were randomly sacrificed on the 16th day by cervical dislocation. Sterile samples of the blood, liver, heart, lungs, kidney, and spleen were then taken, and the organ-to-body mass indexes were computed.

For 48 h at 37 °C, 50 µL of blood was inoculated onto brain–heart infusion (BHI) agar and MRS agar plates. After homogenizing tissue samples in 1 g/mL of buffered peptone water, 100 µL samples of the homogenates were cultured under MRS and BHI agar plates using the identical conditions as blood. The following formula was used to determine the translocation rate [14]:$$Translocation\;Rate=\frac{Number\;of\;mice\;with\;translocation}{Total\;number\;of\;mice}$$

### 2.3. Whole-Genome Sequencing and Analysis of L. reuteri TPC32

The genomic study of *L. reuteri* TPC32 involved the extraction of whole-genome DNA using the Qiagen DNA extraction kit from Qiagen, Antwerpen, Belgium, following the directions provided by the manufacturer. The purity and integrity of the entire DNA sample were assessed by 0.75% agarose gel electrophoresis. The genomic DNA of *L. reuteri* TPC32 was sequenced by Bioyigene Biotechnology Co., Ltd., located in Wuhan, China. The process involved DNA damage repair and terminal repair magnetic bead purification to join and purify the DNA. The Qubit library was then measured and produced using the ligation sequencing kit (SQK-LSK109 Ligation Sequencing Kit, Oxford Nanopore Technologies, Oxford, UK). First, the library was put onto the R9.4 sequencing chip. Then, the PromethION sequencer (Oxford Nanopore Technologies, Oxford, UK) was used to sequence it for 48–72 h.

The data were mixed with a Unicycle, corrected with Pilon or NextPolish and combined with second-generation sequencing data to obtain the final genome sequence. The coding genes were predicted with Prodigal, retaining the complete CDS; tRNA were predicted by tRNAscan-SE (http://lowelab.ucsc.edu/tRNAscan-SE/ accessed on 18 March 2023) [19], and the rRNA genes were predicted by RNAmmer version 1.2 (http://www.cbs.dtu.dk/services/RNAmmer/ accessed on 18 March 2023) [20]. Other ncRNAs were searched using the Rfamv database [21] with infernal for prediction, retaining the predicted length > 80% of the sequence length. The gene islands were predicted with IslandPath-DIMOB (http://pathogenomics.sfu.ca/islandviewer/ accessed on 19 March 2023), and the prophages were predicted with the PhiSpy algorithm [22]. After extraction of genome-encoded proteins, functional annotation is performed using databases including TIGRFAMs (http://www.jcvi.org/cgi-bin/tigrfams/index.cgi/ accessed on 18 March 2023) [23], Pfam (Pfam version 34.0) [24], SwissProt, KEGG (http://www.genome.jp/keg/ accessed on 19 March 2023), RefSeq (NCBI annotation release 104) [25], GO, and COG (https://www.ncbi.nlm.nih.gov/COG/ accessed on 20 March 2023) [26]. The phylogenetic tree of *Lactobacillus* species was built using Orthofinder (https://github.com/davidemms/OrthoFinder/ accessed on 10 January 2024) and drawn using the online software ITOL (Interaction Tree of Life, version 6.0).

### 2.4. Preventive Effects of the Strains against Bacterial Diseases in Mice

The method of purchasing and management for KM (Kunming) female mice was the same as described before. Specifically, mice were orally gavaged with 400 µL of either *L. reuteri* TPC32 (1 × 10^9^ CFU/mL) or sterile saline (vehicle) for two weeks (14 days). Then, the mice bacterial infection model was developed with *S. Typhimurium* (1 × 10^8^ CFU/day) for three days. On the 17th day, the serum separated from whole blood, and the small intestinal tissue was thoroughly processed into homogenate from euthanized mice. Using an ELISA kit (Meimian, Yancheng, China), the blood immunoglobulin levels (IgA, IgG, and IgM), as well as the intestinal secretion of sIgA, TGF-β, IFN-γ, TNF-α, IL-6, and IL-18, were assessed. Next, the newly cut, one-centimeter-long small intestine segments were kept in 10% (*w*/*v*) paraformaldehyde (pH 7.0) for H&E staining, crypt depth analysis, and histological pathology (Image-Pro Plus 6.0). To provide standardization throughout the study, control animals not receiving probiotics were referred to as the “vehicle control group (N group)” and the “infection group (NS group)”, and the protection group was gavaged with *L. reuteri* TPC32 as the “TP group”. The fresh feces were aseptically collected from all three groups to analyze intestinal microbiota structure (*n* = 5) by following our previously published study [11].

### 2.5. Statistical Analysis

Statistical analysis was performed using GraphPad Prism (GraphPad Software, version 8.0). Duncan’s test was used after a one-way analysis of variance to undertake a statistical analysis of the data for multiple comparisons. The threshold for statistical significance was kept at *p* < 0.05.

## 3. Results

### 3.1. Isolation and Characterization of L. reuteri TPC32

Four colonies were selected as potential LAB, and the calcium-dissolving zone/Gram-positive/catalase-negative/was selected. In addition, 16S rRNA sequence analysis showed that *L. reuteri* TPC32 was 99.72% homologous to *L. reuteri* (NR_075036.1), TPC21 was 100% homologous to *Streptococcus alactolyticus* (NR_041781.1), TPK11 was 99.79% homologous to *L. reuteri* (NR_113820.1), and TPM33 was 98.77% homologous to *Enterococcus faecium* (NR_114742.1). Meanwhile, the acid/bile tolerance test showed that the *L. reuteri* TPC32 survival rate was 40.9% (at pH 3.0) and 70.25% (bile salt at 0.3% concentration) combining better growth trends (Figure 1C,D). The zone of inhibition was more than 10 mm, which indicates the bacteriostatic effect of LAB [27]. Our results show four strains with more than 10 mm zone of inhibition (Figure 1E–G). The strain *L. reuteri* TPC32 was selected as the final probiotic LAB for safety assessment.

### 3.2. Safety Assessment of L. reuteri TPC32

As shown in Figure 2A, there was nearly no hemolysis reaction in *L. reuteri* TPC32. All mice remained healthy without any clinical signs, and there was no significant difference in the animal feeding behavior and mental status among the treated and control groups. Subsequently, there were no substantial changes in the body weights of the different groups (Figure 2B,C). The internal organs and intestinal segments were normal without macroscopic ulcer or adhesion. There was no statistical difference in the incidence of translocation of bacteria to the liver or kidney between the control and any of the experimentally treated groups.

### 3.3. Whole-Genome Sequencing and Analysis of L. reuteri TPC32

Sequencing and de novo assembly results showed that the genome length of *L. reuteri* TPC32 was 2,214,495 bp, and the guanine–cytosine content was 38.81%. In addition, the *L. reuteri* TPC32 genome contained a trio of plasmids. A total of 2212 CDS were spotted, averaging a length of 1,921,512 bp, collectively covering 86.77% of the genome. In addition, the chromosome contained 75 tRNA genes, 21 rRNA genes, and 8 genomic islands, and 3 prophages were found in the whole genomes. This whole-genome sequence has been deposited at GenBank under Accession No. PRJNA1065120. The strain genomic features of *L. reuteri* TPC32 are summarized in Appendix A and graphically represented in Figure 3A.

Considering other databases, 2212 proteins were functionally annotated from COG and KEGG databases. For COG functional annotation, 1697 genes (76.72%) were annotated to the 23 COG categories (Figure 3B), which included 6 main functional groups: ribosomal structure and biogenesis (190 genes), prophages/transposons (147 genes), amino acid transport and metabolism (138 genes), transcription (130 genes), carbohydrate transport and metabolism (118 genes), and coenzyme transport and metabolism (115 genes). Genes (1272) were categorized into 6 large classes and 37 subclasses based on KEGG categorization (Figure 3C), including carbohydrate metabolism (118 genes), metabolism of cofactors and vitamins (101 genes), amino acid metabolism (83 genes), translation (81 genes), nucleotide metabolism (70 genes), membrane transport (58 genes), and signal transduction (50 genes).

Subsequently, the 306 genes were categorized into seven functional classifications on CAZyme categorization (Figure 3D). Among those genes, 152 genes were annotated as glycoside hydrolases (GHs), 131 genes were annotated as glycosyl transferases (GTs), 29 genes were annotated as carbohydrate-binding modules (CBMs), 19 genes were annotated as carbohydrate esterases (CEs), 2 genes were annotated as polysaccharide lyases (PLs), 2 genes were annotated as auxiliary activities (AAs), and 1 gene was annotated as the Dockerin domain. Similarly, the genes were annotated using the CARD and VFDB, according to which dalfopristin (*vatE*) and tetracycline (*tetW*) drug-resistance genes were annotated in *L. reuteri* TPC32, whereas virulence genes were not annotated.

Most prominently, the *L. reuteri* TPC32 strains were identified as members of the genus *Lactobacillus*, forming a subcluster in the *L. fermentum* IFO 3956 phylogenetic group, and were highly related to *L. reuteri* 2010, as determined by complete genomes (≥99% sequence similarity). Therefore, the results provide further proof of the strain identified as *L. reuteri* by the phylogenetic tree (Figure 3E).

### 3.4. Preventive Effects of the L. reuteri TPC32 against Bacterial Diseases in Mice

During the initial two weeks, there were no notable alterations in the physical and behavioral characteristics of the mice. However, following the encounter with *S. Typhimurium* three days later, the NS group exhibited symptoms such as tremors, diminished movement, unkempt fur, shut eyes, and congregated behavior, contrasting with the behaviors of the N and TP groups. Notably, a substantial decline in body weight was evident in the NS group, reaching statistical significance (*p* < 0.01) (as shown in Figure 4A). Moreover, the liver-to-body weight ratio in the NS group was markedly elevated compared to the N and TP groups, at a significance level of 0.01. At the same time, the ratio of lung weight to body weight was also notably higher than that of the N group (*p* < 0.05) (refer to Figure 4B).

The analysis of serum immunoglobulins revealed no significant differences in IgA and IgG levels between the TP group and the other groups (*p* > 0.05). However, it was evident that the IgM levels in the TP group were notably decreased in comparison to those in the NS group (*p* < 0.001) (Figure 4C). In the duodenum, the levels of five cytokines were downregulated in the TP group compared with the NS group, in which IFN-γ (*p* < 0.05) (Figure 4D), IL-18 (*p* < 0.05) (Figure 4F), TGF-β (*p* < 0.05) (H), and TNF-α (*p* < 0.05) (Figure 4I) had more significant differences. In the jejunum, the level of IL-6 (*p* < 0.01) (Figure 4E) of the TP group was significantly lower than that in the NS group, whereas the IL-18 level in the TP group was higher than that in the N group (*p* < 0.05). Furthermore, in the ileum, the NS group had significantly higher levels of IFN-γ (*p* < 0.05), IL-6 (*p* < 0.05), and IL-18 (*p* < 0.001 and *p* < 0.05) than the N and TP groups. Interestingly, in the ileum, jejunum, and duodenum, the TP group enhanced sIgA expression (Figure 4G).

The morphological structures of the small intestine in the control group exhibited regular histological features. While in the infected group, there was an infiltration of mucosal inflammatory cells, and the intestinal villus epithelial cells were degenerated and necrotized in the case of the TP group, pathological changes were significantly reduced (Figure 5A–C). Moreover, Figure 5D demonstrates that the TP group’s duodenal mucosal thickness was more significant than that of the NS group (*p* < 0.01). Further, compared to the NS group (*p* < 0.01), the TP group showed neatly and tightly packed columnar epithelial cells with a considerable increase in the villus height–crypt depth ratio (V/C ratio) (Figure 5E).

The Illumina sequencing method yielded high-fidelity sequences from 15 stool samples. The analysis showed that there were no discernible variations between the groups in the intestinal microbiota’s alpha diversity, as measured by the Chao1, Faith_pd, Goods_coverage, Shannon, Simpson, Pielou’s_e, and Observed_species (Figure 6A). Exceptionally, the overall structure of the gut microbiota was significantly shifted in the three groups, as shown by principal coordinate analysis (PCoA) based on unweighted_unifrac distance (Figure 6B).

The taxonomic composition analysis revealed that Bacteroidetes, Firmicutes, and Proteobacteria were the dominant phyla in all samples. Bacteroidetes accounted for 71.4 ± 9% in the N group, 60.6 ± 5.9% in the NS group, and 35.5 ± 16.3% in the TP group. Firmicutes represented 25.9 ± 8.7% in the N group, 37.2 ± 6.0% in the NS group, and 59.4 ± 15.1% in the TP group. Proteobacteria were 1.2 ± 0.5% in the N group, 0.8 ± 0.2% in the NS group, and 1.3 ± 0.8% in the TP group. Other phyla had a lower abundance, accounting for less than 0.5% of all samples (Figure 6C). Similarly, the bacterial genera *Lactobacillus*, *Bacteroides*, [*Prevotella*], *Oscillospira*, *CF*231, *Prevotella*, and *Allobaculum* were shown to be the most abundant (Figure 6D). The primary focus of the analysis was to examine the prevalence of dominating fecal bacteria within the three groups. The TP group exhibited a substantial increase in the relative abundances of *Lactobacillus* compared to the N group (*p* < 0.05) (Figure 6E). Conversely, the N group had a significantly greater abundance of Bacteroides compared to the other groups (*p* < 0.01) (Figure 6E) at the genus level. These data suggest that *S. Typhimurium* altered the prevalence of specific bacterial groups in the gastrointestinal tract. Based on the general development direction, we observed a shift toward restoring normobiosis among the dominant genera following the delivery of intragastric *L. reuteri* TPC32 (Figure 6E).

## 4. Discussion

Probiotics are nonpathogenic microorganisms extracted naturally from gut microbiota and are biologically characterized with potential health benefits, such as the ability to produce digestive enzymes [28], antioxidant and antibacterial properties [29]. They can tolerate bile salts and gastric acids [30], allowing them to thrive and multiply throughout the gastrointestinal system [31]. Therefore, *Lactobacillus* strains were isolated from the feces of healthy Tibetan pigs, and their phenotypic traits were assessed in this study. Four colonies were isolated as potential *Lactobacillus* probiotics and identified with biochemical molecular identification using 16S rRNA sequencing. Likewise, the crucial factor for probiotics to enhance the host’s physiological well-being is their capacity to endure in the intestinal environment [32]. The highly tolerant strain with excellent in vitro antibacterial effects was selected for further research. More importantly, the genomes of *L. reuteri* TPC32 are challenging to differentiate based on the similarity of their 16S rRNA gene sequences. To fully elucidate the genetic structure of metagenomes and gain a more comprehensive understanding of their core genome, specific genes, functional genomics, and evolutionary history, it is imperative to have access to a more significant number of whole genomes. Upon performing a thorough examination of the *L. reuteri* TPC32 genome, we discovered the existence of many genes that play a crucial role in enhancing the effectiveness of probiotics. Genome sequence analysis indicates that the genome length of *L. reuteri* TPC32 was 2,214,495 bp. The guanine–cytosine content was 38.81%. Compared with *L. reuteri* JCM 1112T, *L. fermentum* IFO 3956 [33] and *L. reuteri* KUB-AC5 [34], *L. reuteri* TPC32 had a more extended genome size and lower guanine–cytosine content. Not surprisingly, the genomes with high guanine–cytosine content generate fewer mutations than those with low guanine–cytosine content [35]. This suggests a high degree of variation in the genome of *L. reuteri TPC32*. Intriguingly, these mutations in *L. reuteri TPC32* may make it easier to host adaptation; this property could be beneficial for later application.

Furthermore, these 2,212 genes were functionally annotated by COG, KEGG, and CAZyme. This study categorized a total of 118 genes involved in carbohydrate transport and metabolism, annotated in COG and KEGG pathway, and 13.83% of the genes were associated with carbohydrate-active enzyme (CAZyme). Well-known carbohydrates are metabolized by LAB into short-chain fatty acids (SCFAs) to be used as an additional energy source [36,37]. SCFAs have a virtual role in gut health [38] and microbiota–gut–brain crosstalk [39]. Because of the versatile capability of *L. reuteri* TPC32, it could be used as a health promoter. Several studies have revealed that LAB prevent the symptoms caused by vitamin B deficiency during pregnancy [40] and produce vitamin K during fermentation [41]. The present study suggests that a high proportion of genes were assigned to the metabolism of cofactors and vitamins. This illustrates that the *L. reuteri* TPC32 strain contributes to the biotechnological production of vitamin supplements and strategies for preventing vitamin deficiencies. Unfortunately, various case reports of probiotic-associated bacteremia, fungemia, endocarditis, liver abscess, and pneumonia have been published, even though the ingested probiotics are known to possess low-virulent and nonpathogenic properties [42]. To investigate further, we analyzed the virulence genes of *L. reuteri* TPC32 using VFDB and safety assessment studies in mice. The current study detected no demonstrable abnormality in mice after two weeks of feeding with the test probiotic *L. reuteri* TPC32 and no hemolysis reaction. Thus, these results indicate that *L. reuteri* TPC32 has no adverse effect on the general health status of mice. Moreover, the resistance to antibiotics is a vital aspect of a potential probiotic strain in safety assessment. The antibiotic-resistance genes of glycopeptides, sulfonamides, penicillin, aminoglycosides, and cephalosporins have already been discovered in LAB [43], while *tetW* has been known as a highly prevalent resistance gene [44]. This study found that *vatE* and *tetW* genes were annotated in *L. reuteri TPC32,* whereas conjugal transfer needs further verification.

Despite this, numerous studies have revealed that LAB are widely used for fermentation dairy products [45] and anticancer activities [46]. They also regulate immunity [47] and antioxidant properties [13], and a current study reveals the prebiotic properties of *L. reuteri* TPC32 in an in vitro experiment. Nonetheless, whether *L. reuteri* TPC32 can improve immunity to protect against *S. Typhimurium* damage in mice is unknown. Therefore, we investigated the protective effects of *L. reuteri* TPC32 for mice infected with *S. Typhimurium* from blood immune indicators, intestinal cytokines, the intestinal physical barrier, and the intestinal microbiota structure. We found that *L. reuteri* TPC32 can improve mice’s health in many ways. On antigen stimulation, memory cells proliferate and differentiate into plasma cells, which produce glycoprotein (i.e., immunoglobulin) against the antigens [48]. The IgG, IgA, and IgM levels were significantly elevated in *S. Typhimurium*-infected mice. Fortunately, serum IgM levels dropped significantly with *L. reuteri* TPC32, meaning that the immune system fights infections by enhancing humoral immunity in mice. Secretory IgA (SIgA) is the predominant immunoglobulin on mucosal surfaces. It serves as a formidable barrier against bacterial and viral pathogens [49], while previous research has shown that it reduces the invasion of enterica with *Salmonella* [50]. Similarly, *L. reuteri* TPC32 can enhance the secretion of intestinal SIgA to a certain degree. These findings align with a recent study that showed the capacity of genetically modified LAB to stimulate the synthesis of SIgA in the intestines [51]. Accordingly, *L. reuteri* TPC32 has an increasing effect on the SIgA level to alleviate intestinal inflammation by inhibiting pathogen adhesion to the mucosal surface and clearing them. In general, pathogenic microbes cause damage to the intestinal mucosa, which then activates submucosal lamina propria macrophages and T lymphocytes to secrete a significant quantity of cytokines, such as IL-1, IL-6, IL-17, IL-23, and TNF-α [52]. Meanwhile, another example elicited that LAB affect intestinal cytokine and reduce the cytokine storm to improve gut health [53]. Remarkably, our results suggest a general principle by which *L. reuteri* TPC32 therapy reduced the enteric inflammatory factors IL-6, IL-18, TNF-α, and TGF-β in *Salmonella*-infected mice.

The small intestine is the primary location where gastrointestinal tract nutrition absorption mainly occurs, as well as an important line of defense against commensal GI microbiota. Admittedly, intestinal stem cells (ISCs) in crypts undergo differentiation into many types of epithelial cells, such as intestinal stem cells, Paneth cells, Goblet cells, neuroendocrine cells, and intestinal epithelial cells. Additionally, they play a crucial role in maintaining balance and regenerating the epithelial cells of the small intestine [54,55]. Therefore, intestinal mucosal integrity is considered a good marker of intestinal health. In particular, villus height, crypt depth, and V/C ratio are commonly used metrics to evaluate the structure of the intestine [56]. The findings of our study indicate that *L. reuteri* TPC32 effectively slowed down the progression of pathological damage and preserved the average depth of the crypts to prevent crypt hyperplasia. Additionally, the V/C ratio in mice showed a considerable rise. The ISC niche has two primary constituents: the extracellular matrix (ECM) and the cellular microenvironment. Mechanistically, the cellular microenvironment consists of all the cells (intestinal mucosal epithelial cells) surrounded by the extracellular matrix (ECM). These cells produce a variety of matrix components and mucosal immune factors to control the self-renewal and differentiation of intestinal stem cells (ISCs) [57]. Considering recent research, our results provide significant information regarding the relationship between physical barriers and immunological barriers in regulating intestinal barrier function. On the other hand, the intestinal niche of ISCs is exposed to a remarkable abundance and variety of bacteria and constitutes the intestinal microbial mucosal barrier, and the gut microbiota has a role in the development and stimulation of both the mucosal and systemic immune systems [58]. Alternatively, the imbalance in the composition and quantity of intestinal microorganisms causes the colonization of pathogenic bacteria, consequently affecting intestinal permeability. Unquestionably, the intestinal microbiota structure and metabolism were disordered when exposed to *S. typhimurium* [59].

Interestingly, increasing evidence suggests that LAB ameliorate autoimmune neuritis, have a regulatory effect on antibiotic-associated diarrhea, improve nitrogen metabolism in weaned piglets, and alleviate enterohaemorrhagic *Escherichia coli*-induced diarrhea [60] by regulating intestinal microbiota composition. Our current study shows that Bacteroidetes, Firmicutes, and Proteobacteria were the most abundant bacterial phyla in all three groups. However, the level of Firmicutes in the TP group significantly increased compared to the other groups. This rise may be attributed to the supplementation of *L. reuteri* TPC32. Uniformly, the results consistently showed that the abundance of *Lactobacillus* significantly increased after the oral administration of *L. reuteri* TPC32, as determined by species composition difference analysis. *Lactobacillus* is a beneficial bacterium that helps maintain a healthy balance of microorganisms in the intestines and protects against the invasion of harmful pathogens [60,61,62]. As shown in Figure 6E, the remaining prevailing genera did not exhibit any substantial variation among the three groups; nevertheless, the intestinal microbiota displayed a propensity to revert to its natural state. Our hypothesis suggests that *L. reuteri* TPC32 maintained the normal gut microbiota and enhanced the population of beneficial bacteria, resulting in a positive regulatory effect on intestinal health in mice. There is no denying that this study still has many shortcomings. Due to data limitations, this study is limited by a small sample size and a short research duration.

## 5. Conclusions

Overall, our study demonstrates that *L. reuteri* TPC32 has the potential to be a probiotic, as it was found to survive and exhibited probiotic properties at a genetic level. *L. reuteri* TPC32 also facilitated the restoration of the intestinal microbiota, bringing it closer to a healthy intestinal microbiota composition. This regulatory mechanism causes the ISC niche to survive in a typical intestinal microecological environment. It strengthens the intestinal immune barrier by reducing the colonization of intestinal pathogenic bacteria and decreasing intestinal mucosal damage. Thus, this study reveals that intestinal microbiota, such as *L. reuteri* TPC32, could modulate intestinal physical, immune, and microbial barriers against systemic disease in *S. Typhimurium*-infected mice.

## Figures and Tables

**Figure 1 nutrients-16-01900-f001:**
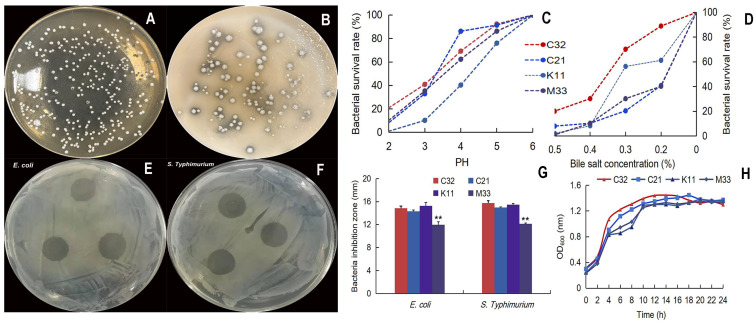
Isolation and characterization of *L. reuteri* TPC32: (**A**) morphology of MRS agar medium; (**B**) the calcium-dissolving zone of LBA; (**C**) the tolerance of the isolated strains to acid; (**D**) the tolerance of the isolated strains to bile salts; (**E**) inhibition zones of *E. coli* by *L. reuteri* TPC32; (**F**) inhibition zones of *S. Typhimurium* by *L. reuteri* TPC32; (**G**) the results of the antibacterial experiment of *L. reuteri* TPC32; (**H**) the growth curve of LAB broth. One-way ANOVA was usexamine the results. Every piece of data shows the means ± SD; ** *p* < 0.01; unmarked indicates no significant change (*p* > 0.05).

**Figure 2 nutrients-16-01900-f002:**
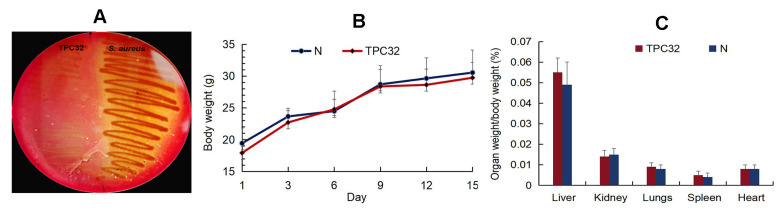
The safety assessment of *L. reuteri* TPC32: (**A**) the hemolysis test of *L. reuteri* TPC32; (**B**) body weight of mice after *L. reuteri* TPC32 feeding; (**C**) organ weight/body weight of mice after *L. reuteri* TPC32 feeding.

**Figure 3 nutrients-16-01900-f003:**
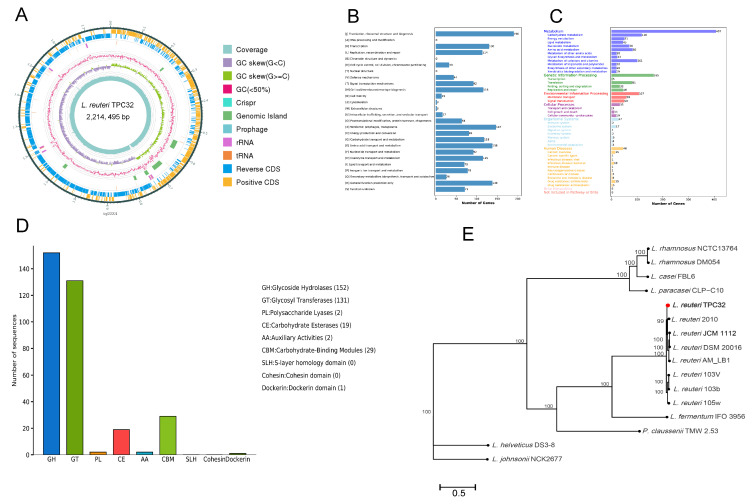
Whole-genome sequencing and analysis of *L. reuteri* TPC32: (**A**) circular genome mapping of *L. reuteri* TPC32. Marked information is displayed from the outer circle to the innermost as follows: CDS on the forward strand, CDS on the reverse strand, tRNA and rRNA, GRISPR, prophage regions and genomic islands, GC content, GC skew, and sequencing depth. The number of genes assigned in COG (**B**) and KEGG (**C**) categories; (**D**) annotation of the carbohydrate-active enzyme of *L. reuteri* TPC32; (**E**) a phylogenetic tree was reconstructed based on whole-genome sequences; the bootstrap-support value before each node represents the confidence degree of each branch.

**Figure 4 nutrients-16-01900-f004:**
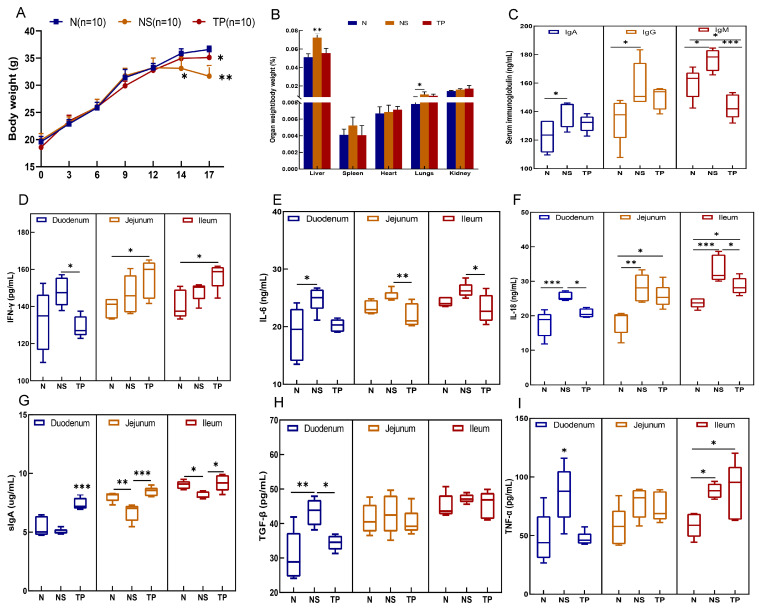
Preventive effects of the *L. reuteri* TPC32 against bacterial diseases in mice: (**A**) body weight; (**B**) organ-to-body weight ratio; (**C**) serum immunoglobulins in mice; (**D**) IFN-γ; (**E**) IL-6; (**F**) IL-18; (**G**) sIgA; (**H**) TGF-β; (**I**) TNF-α. One-way ANOVA was used to examine the results. All data represent means ± SD. Unmarked indicate no significant difference (*p* > 0.05); * *p* < 0.05; ** *p* < 0.01; *** *p* < 0.001.

**Figure 5 nutrients-16-01900-f005:**
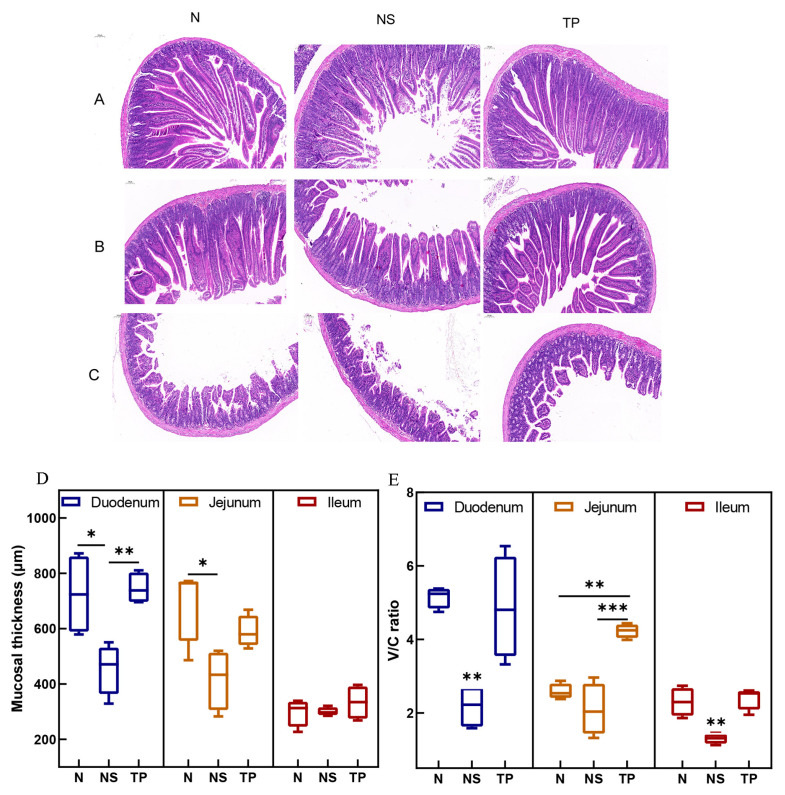
Effect of *L. reuteri* TPC32 on the function of the intestinal physical barrier in mice infected with *S. Typhimurium*: (**A**–**C**) H&E-stained histological analysis of the ileum, jejunum, and duodenum on a 100 μm scale bar; (**D**) the intestinal segments’ mucosal thickness; (**E**) the proportion of crypt depth to small intestine villus height. One-way ANOVA was used to examine the results. Every piece of data shows the means ± SD. Unmarked indicate no significant difference (*p* > 0.05); * *p* < 0.05; ** *p* < 0.01; *** *p* < 0.001.

**Figure 6 nutrients-16-01900-f006:**
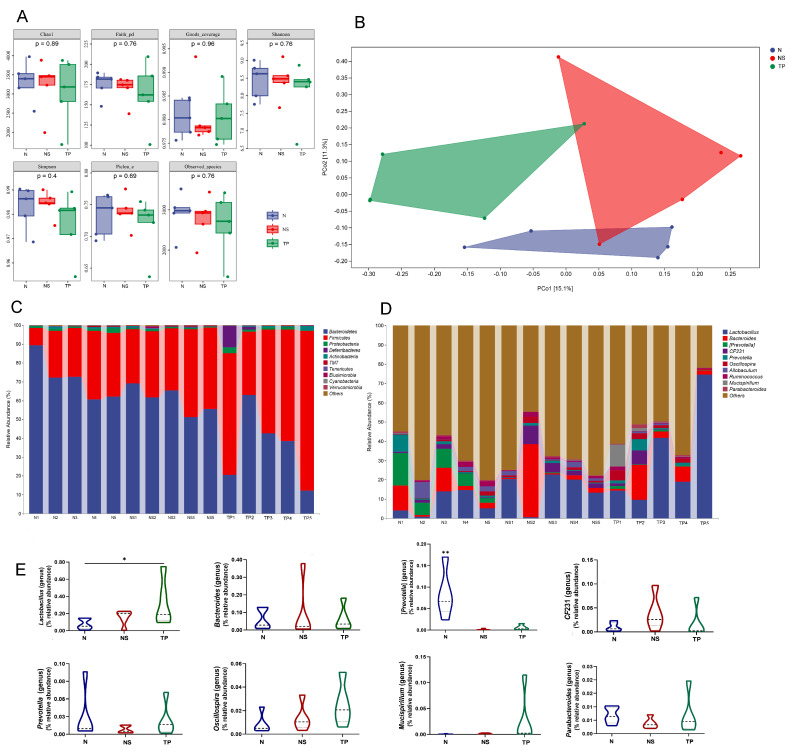
*L. reuteri* TPC32 influences the composition of the gut microbiota: (**A**) the alpha diversity of the fecal microbiota was assessed using various diversity indices, including Chao1, Faith_pd, Goods_coverage, Shannon, Simpson, Pielou’s_e, and Observed_species; (**B**) analysis of the microbial community in fecal samples using principal component analysis; the provided PCA map was generated using the Euclidean distance metric. Each point on the map represents a single sample, and the distance between the two points reflects the dissimilarity in fecal microbiota. The 16S rRNA gene sequencing revealed the relative abundance of intestinal microbiota in different groups at the (**C**) phylum and (**D**) genus levels; (**E**) there were substantial differences in the abundance of certain species at the genus level. The results were assessed using one-way ANOVA. The results presented in the study are shown as means ± standard deviation (SD). * *p* < 0.05; ** *p* < 0.01.

## Data Availability

All data that were generated or analyzed during this study are included in this published article. This study’s whole-genome sequence and 16S rRNA datasets can be found in online repositories. The names of the repository/repositories and accession number(s) can be found at: https://www.ncbi.nlm.nih.gov/. The NCBI login numbers are PRJNA1065120, PRJNA880707, and PRJNA880707.

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
