# Peer review of "The Whole-Genome Sequencing and Probiotic Profiling of Lactobacillus reuteri Strain TPC32 Isolated from Tibetan Pig"

_nutrients, 2024, doi:10.3390/nu16121900_

Round 1

Reviewer 1 Report

Comments and Suggestions for Authors

The paper is written with great skill and the conceptualization and methodology of the experiment are commendable. Congratulations to the authors for their hard work and dedication.

Some suggestions to improve the quality of the paper:

In the Statistical analysis, please include how the assessment of required mice in each group was assessed.

Add strengths and limitations of the study.

The graphical abstract of the study or experiment/isolation of bacteria will be interesting. Please consider.

Some less important issues:

In the abstract section, basic information about the method should be included.

[14]and; respectively[19]. – space

[17]. m – m should be replaced by M (line 66), as same as A by a „membrane. alternatively” (line 67)

Author Response

Dear Editor and Referees:

Thoughts for your letter and for the editors’ and referees’ comments concerning our manuscript entitled “Complete Genome Sequencing and Probiotic Profiling of Lactobacillus Reuteri Strain TPC32, Isolated from Tibetan Pig”. Those comments are all valuable and very helpful for revising and improving our paper, as well as the important guiding significance to our researches. We have studied comment carefully and have made correction which we hope meet with approval. Revised portion are marked in red in the paper. The main corrections in the paper and the responds to editors’ and referees’ comments are as flowing:

Responds to the editors and reviewers:

  1. Response to comment: (In the abstract section, basic information about the method should be included.)

Response: This section of the article has been edited and marked in red type.

  1. Response to comment: (In the Statistical analysis, please include how the assessment of required mice in each group was assessed.)

Response: The explanation for this matter can be found on line 205 of the article. The study involved a total of 10 mice, as three replicates were considered biologically meaningful. Hence, this investigation utilized the slaughter and 16s RNA sequencing of five mice.

  1. Response to comment: (Add strengths and limitations of the study.)

Response: Critical detection methods have been added to the abstract of the article.

  1. Response to comment: ([14]and; respectively[19]. – )

Response: This section of the article has been edited and marked in red type.

  1. Response to comment: (m – m should be replaced by M (line 66), as same as A by a „ alternatively” (line 67).)

Response: This section of the article has been edited and marked in red type.

Special thanks to you for your.

Sincerely,

Qinghui Kong.

Reviewer 2 Report

Comments and Suggestions for Authors

The manuscript describes the characterization of a potentially probiotic lactic acid bacteria strain, with a fairly extensive molecular and functional description. In general, in view of the results provided, it can be considered that this strain does meet the usual criteria to be considered probiotic, at least at the doses investigated to prevent Salmonella Typhimurium infection.

However, the manuscript is not homogeneous throughout, since while the results, discussion and conclusions seem to be described and written in a systematic, coherent and careful way, the first parts seem to have been elaborated by a different person, with a much less careful style of writing, formatting and grammar.

All these aspects must be improved in an important way prior to the final acceptance of the manuscript.

Specific comments:

In the abstract section, there are several abbreviations, such as “GC”, “CARD”, “VFDB” that are not defined. Moreover, they appear only once in the text, so it does not make sense that they are abbreviated. Similarly, all the names of bacterial genera and species are not written in italics, and the first time they appear in the text. “L. reuteri” should have been cited as “Lactobacillus reuteri”.

Page 1, line 41: "lactic acid bacteria" was previosly abreviated as "LAB"

Page 2, line 48: "CNS" should be defined.

In general, all the introduction sections is difficult to read and somewhat disjointed. It needs to be completely restructured.

page 2, lines 46-54: "melamona", "tumor microenvironment"...what is the relation between this paragraph and the current article? I believe this paragraph does not contribute anything relevant to this article.

In all manuscript, the term "flora" is widely used. It sould be better if this term is changed by "microbiota".

Page 2, lines 88-89: in the heading of the sections, it should be clarified that “TPC32” corresponds to Lactobacillus reuteri

Page 2, line 122 and througouth the manscript. The names of bacterial species all written in several cases whout italics, and in some cases the correct format, with the entire name the firts time that apppears in the text and the abbreviated name afterwards was not followed. Please, correct it.

Page 2, line 140: "Unified credit code" should be deleted.

Page 3, line 142: Why were chosen a dosage of 1x108 CFU/day for 15 consecutive days?

Page 4, line 186: "CMCC" should be defined.

Page 5, lines 200-203: What software was used for statitical analysis?

211-212: "

Page  5, lines "To identify the potential pro-211 biotic LAB candidates, these strains were tested for acid/bile tolerance properties and in 212 vitro bacteriostatic properties". This is materials and methods, and should be placed in the own section. It is also apllicable for other paragraps, such as page 9, "To analyze the alteration in the intestinal microbiota structure in mice protective 323 against S. Typhimurium infection for TPC32, the Illumina sequencing system was used to 324 generate high-quality sequences from 15 fecal samples."

Figures 3 and 6 are completely unreadable. If a figure appears in the text, it must be readable, otherwise its presence is meaningless. In this case, such figures should be divided, enlarged or even eliminated if the authors consider it necessary, but if they cannot be read, their presence in the manuscript is meaningless.

In general, the names of bacteria in the references section, as well as the trm "in vivo", "in vitro", and latin expression, should be written in italics.

Comments on the Quality of English Language

The English quality of the abstract, Introduction and Materials and methods section is quite improvable

Author Response

Dear Editor and Referees:

Thoughts for your letter and for the editors’ and referees’ comments concerning our manuscript entitled “Complete Genome Sequencing and Probiotic Profiling of Lactobacillus Reuteri Strain TPC32, Isolated from Tibetan Pig”. Those comments are all valuable and very helpful for revising and improving our paper, as well as the important guiding significance to our researches. We have studied comment carefully and have made correction which we hope meet with approval. Revised portion are marked in red in the paper. The main corrections in the paper and the responds to editors’ and referees’ comments are as flowing:

Responds to the editors and reviewers:

  1. Response to comment: (In the abstract section, about“GC”, “CARD”, “VFDB”and “ reuteri”)

Response: The abbreviation GC has been revised to represent guanine-cytosine. Similarly, CARD has been amended to stand for 'Comprehensive Antibiotic Research Database'. Additionally, the abbreviation VFDB has been altered to represent Virulence factor database in the text, and this modification is indicated by the use of red font.

  1. Response to comment: (Page 1, line 41: "lactic acid bacteria" was previosly abreviated as "LAB")

Response: Modifications were implemented to the introduction sections, and the revised part has been marked in red.

  1. Response to comment: (Page 2, line 48: "CNS" should be defined.In general, all the introduction sections is difficult to read and somewhat disjointed. It needs to be completely restructured.)

Response: Modifications have been implemented in the Introduction, and the relevant material has been removed.

  1. Response to comment: (About page 2, lines 46-54)

Response: This section of the article has been revised and removed.

  1. Response to comment: (In all manuscript, the term "flora" is widely used. It sould be better if this term is changed by "microbiota".)

Response: The article has substituted the term "flora" with the more precise term "microbiota".

  1. Response to comment: (Page 2, lines 88-89: in the heading of the sections, it should be clarified that “TPC32” corresponds to Lactobacillus reuteri.)

Response: Lactobacillus reuteri TPC32 has been designated as Lactobacillus on page 1, line 15. L. reuteriTPC32, and changes have been made everywhere in the article.

  1. Response to comment: (About Page 2, line 122 and througouth the manscript.........)

Response: The item indicated in red on lines 129, 130, and 139 provides the defined abbreviations for bacteria.

  1. Response to comment: (Page 2, line 140: "Unified credit code" should be deleted. )

Response: This part has been deleted from the article.

  1. Response to comment: (Page 3, line 142: Why were chosen a dosage of 1x108CFU/day for 15 consecutive days? )

Response: There is evidence indicating that consuming Lactobacillus reuteri (1x108) for a duration of 14 days can potentially modify the composition of the gut microbiota, control the immune system, and offer protection to the gastrointestinal tract. Furthermore, the article referenced this literature. The specific references are as outlined below: 

[1] Liu, H. Y.; Gu, F.; Zhu, C.; Yuan, L.; Zhu, C.; Zhu, M.; Yao, J.; Hu, P.; Zhang, Y.; Dicksved, J., et al. Epithelial Heat Shock Proteins Mediate the Protective Effects of Limosilactobacillus reuteri in Dextran Sulfate Sodium-Induced Colitis. Front Immunol. 2022, 13, 865982. 

[2] He, B.; Hoang, T. K.; Tian, X.; Taylor, C. M.; Blanchard, E.; Luo, M.; Bhattacharjee, M. B.; Freeborn, J.; Park, S.; Couturier, J., et al. Lactobacillus reuteri Reduces the Severity of Experimental Autoimmune Encephalomyelitis in Mice by Modulating Gut Microbiota. Front Immunol. 2019, 10, 385.

  1. Response to comment: (Page 4, line 186: "CMCC" should be defined.)

Response: CMCC stands for China Medical Bacteria Preservation Management Center, and this section of the article has been altered.

  1. Response to comment: (Page 5, lines 200-203: What software was used for statitical analysis?.)

Response: Statistical analysis was performed using GraphPad Prism (GraphPad Software, v 8.0 ).The article has been revised and the revisions are indicated in red.

  1. Response to comment: (About Page 5, "To identify the potential pro-211 biotic LAB candidates, these strains were tested for acid/bile tolerance properties and in 212 vitro bacteriostatic properties".........)

Response: This section of the article has been edited and marked in red type.

  1. Response to comment: (About Figures 3 and 6)

Response: The photographs in FIG. 3 and FIG. 6 are unreadable due to insufficient resolution. The resolution has been adjusted and the original pictures have been submitted to the editorial department for revision.

  1. Response to comment: (In general, the names of bacteria in the references section, as well as the trm "in vivo", "in vitro", and latin expression, should be written in italics.)

Response: This section of the article has been edited and marked in red type.

Special thanks to you for your.

Sincerely,

Qinghui Kong.

Round 2

Reviewer 2 Report

Comments and Suggestions for Authors

All the questions and comments made by the Reviwer in the first round of review has been satisfactorely answered or changed in the manuscript. The manuscript is now aceptable for its publication.

Author Response

Dear Editor and Referees: Thoughts for your letter and for the editors’ and referees’ comments concerning our manuscript entitled “Complete Genome Sequencing and Probiotic Profiling of Lactobacillus Reuteri Strain TPC32, Isolated from Tibetan Pig”. Those comments are all valuable and very helpful for revising and improving our paper, as well as the important guiding significance to our researches. We have studied comment carefully and have made correction which we hope meet with approval. Revised portion are marked in red in the paper. The main corrections in the paper and the responds to editors’ and referees’ comments are as flowing: Responds to the editors and reviewers: 1.Response to comment: (As mentioned by reviewer 2, some obvious writing and grammar mistakes still remain just by a quick a look, such as the abstract and methods (i.e. line 145-147)) Response: We have made changes to the English grammar of this article and the revised part has been marked in red. 2.Response to comment: (About “Reviewer 1 asked to add strengths and limitations of the study”) Response: This is added to lines 478-479 of the manuscript, and the revised part has been marked in red. Special thanks to you for your. Sincerely, Qinghui Kong.